# Modelling RT-qPCR cycle-threshold using digital PCR data for implementing SARS-CoV-2 viral load studies

Fabio Gentilini[1]*, Maria Elena Turba[2], Francesca Taddei[3], Tommaso Gritti[3], Michela Fantini[3], Giorgio Dirani[3], Vittorio Sambri[3,4]

1 Department of Veterinary Medical Sciences, University of Bologna, Ozzano dell'Emilia, Bologna, Italy,
2 Xenturion srl, Forlì, Italy, 3 Unit of Microbiology, The Great Romagna Hub Laboratory, Pievesestina, Italy,
4 Department of Experimental, Diagnostic and Specialty Medicine - DIMES, University of Bologna, Bologna,
Italy

ᴑ These authors contributed equally to this work.
* fabio.gentilini@unibo.it

**Data Availability Statement:** All relevant data are within the manuscript and its S1 and S2 Files.

**Funding:** This work was supported by the University of Bologna [grant Sambri287] and AUSL Romagna [grant COVdPCR].

## Abstract

### Objectives

To exploit the features of digital PCR for implementing SARS-CoV-2 observational studies by reliably including the viral load factor expressed as copies/μL.

### Methods

A small cohort of 51 Covid-19 positive samples was assessed by both RT-qPCR and digital PCR assays. A linear regression model was built using a training subset, and its accuracy was assessed in the remaining evaluation subset. The model was then used to convert the stored cycle threshold values of a large dataset of 6208 diagnostic samples into copies/μL of SARS-CoV-2. The calculated viral load was used for a single cohort retrospective study. Finally, the cohort was randomly divided into a training set (n = 3095) and an evaluation set (n = 3113) to establish a logistic regression model for predicting case-fatality and to assess its accuracy.

### Results

The model for converting the Ct values into copies/μL was suitably accurate. The calculated viral load over time in the cohort of Covid-19 positive samples showed very low viral loads during the summer inter-epidemic waves in Italy. The calculated viral load along with gender and age allowed building a predictive model of case-fatality probability which showed high specificity (99.0%) and low sensitivity (21.7%) at the optimal threshold which varied by modifying the threshold (i.e. 75% sensitivity and 83.7% specificity). Alternative models including categorised cVL or raw cycle thresholds obtained by the same diagnostic method also gave the same performance.

### Conclusion

The modelling of the cycle threshold values using digital PCR had the potential of fostering studies addressing issues regarding Sars-CoV-2; furthermore, it may allow setting up

**Competing interests:** The authors have declared that no competing interests exist.

predictive tools capable of early identifying those patients at high risk of case-fatality already at diagnosis, irrespective of the diagnostic RT-qPCR platform in use. Depending upon the epidemiological situation, public health authority policies/aims, the resources available and the thresholds used, adequate sensitivity could be achieved with acceptable low specificity.

## Introduction

A year after severe acute respiratory system coronavirus 2 (SARS-CoV-2) was declared to be a pandemic [1], many aspects of the infection still remain undefined. In particular, the role of viral loads (VLs) in infectivity and case-fatality rates is still poorly clarified and scarcely used to implement public health measures [2–9].

Since the beginning, it has been clear that VLs have varied greatly among patients over the course of disease, and that infectivity was associated with higher VLs [5, 8]. With respect to SARS, however, high VLs may also be evident in the pre-symptomatic phase, and the peak of viral shedding was observed early in the course of the disease [2, 7, 9]. Furthermore, the role of the VL in the respiratory tract in predicting mortality is also not well-known, although it was evident that higher VLs were associated with higher case-fatality ratios. One of the main hindrances to assessing VLs lies in the inherent difficulty of absolutely quantifying SARS-CoV-2. In fact, reverse transcription quantitative polymerase chain reaction (RT-qPCR) could provide absolute quantification by using labour-demanding daily calibration procedures which, in turn, require not readily available reference materials [10]. Diagnostic laboratories worldwide have been buried by an impressive demand for diagnostics and have hardly been able to face any additional investigational activity. As a result, the majority of the studies regarding VLs have evaluated the cycle threshold (Ct), automatically calculated by thermal cyclers, as a rough quantitative estimate of VL [4–7, 9, 11–16].

Digital PCR (dPCR) is a straightforward evolution of PCR with some obvious advantages over standard qPCR assays. Specifically, dPCR allows for the absolute quantitation of nucleic acid samples without the need for a calibration curve, thanks to compartmentalization by partitioning of the target nucleic acid in thousands of small volume vessels [17]. Thanks to these features, dPCR is inherently more sensitive, specific and precise than standard qPCR, and is specifically reliable for VL absolute quantification [18, 19]. In the face of its many advantages over RT-qPCR, dPCR has still been limited by much higher costs for analysis and a longer turnaround time (TAT), which restricts its application as ancillary or complementary to RT-qPCR. In fact, to date, many studies have demonstrated the superiority of dPCR when compared to RT-qPCR in terms of diagnostic performance [20–31]. However, studies relying on dPCR have not been based on consistent case numbers and the VLs were quantified in only relatively small cohorts. To date, dPCR has been utilised for investigating SARS-CoV-2 for VL quantification in regard to infectivity [2], and disease course monitoring [32, 33], and as a tool for assessing the circulating RNAaemia as an outcome predictor [34–38], as a diagnostic tool for specifically reducing the false negative results for discharging convalescent patients [27, 30, 39], when inhibition was likely as in examining crude sample lysates or samples without RNA purification [21, 35, 40], or wastewater [41], for analysing contaminated surfaces [39] or for preparing standard material for RT-qPCR or cell cultures [42–45]. Moreover, the overwhelming demand for diagnostic testing and the very low TAT required during the epidemic "waves" were scarcely suited to the majority of dPCR platforms. As a result, the vast majority of Covid-

19 cases, being evaluated with only RT-qPCR or dPCR were generally characterised by relatively low consistency cohorts of cases and had a limited impact on SARS-CoV-2 knowledge.

To overcome this drawback, this study attempted to model the relationship between the Ct and genome absolute quantification for calculating the VL expressed as copies/μL; this was carried out using the stored Ct value retrieved from the medical records of a public centralised diagnostic laboratory intensely involved in diagnosing SARS-CoV-2. The aim was to test the hypothesis that a model built on a small subset of data could be advantageously harnessed to infer the VL in a large cohort. To that aim, the calculated VLs (cVLs) were then investigated in relationship to chronological fluctuations and differences between age groups, or were used to investigate its predictive power for the outcome.

## Materials & methods

### Ethics statement

The study was conducted according to the guidelines of the Declaration of Helsinki, and approved by the Institutional Review Board of AUSL Romagna under the protocol code "COVdPCR of 07/02/2020. The study has been performed using exclusively anonymized, left-over samples deriving from the routine diagnostic procedures therefore the Ethical approval or informed consent is not required. The anonymization was achieved by using the current procedure (AVR-PPC P09, rev.2) checked by the local Ethical Board.

### Experimental layout

The present study was composed of three steps:

1. The first step was aimed at defining a function to convert the Ct values obtained using diagnostic RT-qPCR to absolute quantification as genome copies/μL carried out using dPCR and to assess the respective error. This task was achieved using a linear regression model built using a small cohort of 51 samples.

2. After defining the regression function and its accuracy, the equation was used to calculate the VL in a very large cohort of 6208 Covid-19 cases. The cVLs were investigated in an observational study with a cross-sectional retrospective design.

3. Finally, the medical data, including cVL, was used to build a straightforward predictive model, and its accuracy was calculated in a single cohort retrospective study and compared with a model including the raw Ct value.

### Samples

Digital PCR has been demonstrated to be suitable for the retrospective evaluation of universal transport medium (UTM)-stored SARS-CoV-2 positive samples [31]. On this basis, 51 RNA samples conserved at -80˚C in UTM (Copan, Copan Italia SpA) were selected from all the diagnostic samples examined at the "Great Romagna Hub Laboratory Pievesestina" (AVR Centro Servizi Laboratorio Unico Pievesestina, Cesena) during the Covid-19 pandemic. All the samples had been collected using nasopharyngeal or oropharyngeal swabs (Copan), immediately transferred into tubes containing 3 mL of UTM and transported to the diagnostic laboratory for SARS-CoV-2 testing using one of many different RNA purification platforms and RT-qPCR assays (S1 File). The results were expressed as positive or negative together with the Ct values of the respective targets; some anamnestic, epidemiological and clinical data were retrieved from the Laboratory database. The case-fatality information was recovered from the

Death Registries of the Public Health Departments of the local medical services of Romagna. The samples were retrieved from the repository, the RNA was purified and re-assessed using both RT-qPCR and dPCR. In addition, the samples were divided into two sets; the first set of 13 samples (training set) was used to create the regression model while the second set of 38 samples (evaluation set) was used to validate the model. Regardless of the method originally used, the cohort of 51 samples was composed by stratifying the samples into high ($\leq$ 20 Ct), medium ($> 20 \leq 25$ Ct) and low ($> 25$ Ct) VL categories using the recorded Ct.

### RNA purification, RT-qPCR and dPCR

The samples were collected using oro and nasopharyngeal swabs immediately placed in UTM (Copan, Copan Italia SpA). The RNA was purified from UTM, and used for RT-PCR and dPCR assays. The detailed protocols are reported in S1 File.

### Statistical analysis and modelling

The analytical performances of the dPCR assay were established in terms of analytical sensitivity, precision and linearity, and were expressed as Limit of Detection (LOD) and Coefficient of Variation % (CV%) across technical replicates carried out over different days, and as a linear coefficient of correlation $R^2$, respectively. The analytical performances were evaluated using Analyse-it software (Analyse-it Software, UK). (S1 File).

Fifty-one samples positive at Sars-CoV-2 RT-qPCR were retrieved from the repository and divided into two sets: a training set composed of 13 samples and an evaluation set composed of 38 samples. The training set samples were analysed in triplicate with dPCR, and the findings were included in building the model. After that, the 38 samples of the evaluation set were also assessed in single using dPCR, and the results were used to validate the linear regression model. The dPCR results, expressed in terms of $\log_{10}$ copies/$\mu$L of cDNA, were entered as dependent variables and the Ct values as predictors using STATA v12 software. The software allowed calculating both the fitting of the model as a Pseudo R-squared value and its significance beyond the terms of the linear regression function $Y = aX + b$ where $Y$ is the $\log_{10}$ copies/$\mu$L, $X$ is the $C_t$ measured in the RT-qPCR, $a$ is the coefficient of X as defined by the model and $b$ the constant (Table 1).

The 38 samples of the evaluation set were used to test the model. To that end, the predictor formula was used to calculate the absolute counts using the Ct of each sample of the evaluation group. All the samples in the evaluation group were then assayed once using dPCR, and the results were compared with those obtained using the predictor formula. The accuracy of the

**Table 1. Linear regression model including copies/μL as a dependant variable and cycle threshold (Ct) as a predictor factor.**

| $y = ax+b$ <br> $y = \text{LOG}_{10} \text{ (copies/}\mu\text{L)}$ <br> $x = $ Cycle threshold <br> $a = $ Cycle threshold coefficient <br> $b = $ constant | | | | | |
|---|---|---|---|---|---|
| | coefficient | Robust SE | n | R-squared | Root MSE |
| Log copies | | | 38 | .900 | .454 |
| Ct | -.307 | .018 | | | |
| constant | 10.55 | .431 | | | |
| $\log_{10}AbsQuant = -.307[Ct]+10.55$ | | | | | |

SE: Standard Error; MSE: Mean error sum of squares. R-squared is an indicator of reliability of the model. Root MSE is an indicator of accuracy of the model.

prediction was calculated as the Median Absolute Deviation (MAD) of the percentage error (PE) calculated using the formula PE = Absolute value (measured − calculated/measured) x 100. The formula was used to calculate the VL in a cohort of 6208 cases. The cVL as well as the raw Ct and other medical data including gender, age, presence of signs and symptoms, ward/unit of origin, administrative origin, turnaround-time (urgency), date of sampling, type of sampling (oro or nasopharyngeal swabs) were also entered into a logistic regression model to investigate their role as a predictor of case-fatality (outcome alive or dead). In particular, age and cVL were evaluated either as continuous variables or as factors after categorisation according to the following: age (< 6 years-old, $\geq 6$ and $< 18$; $\geq 18$ and $< 30$; $\geq 30$ and $< 50$; $\geq 50$ and $< 70$; $\geq 70$); cVL ($<1$ copies/µL; $\geq 1$ and $< 10^1$; $\geq 10^1$ and $< 10^2$; $\geq 10^2$ and $< 10^3$; $\geq 10^3$ and $< 10^4$; $\geq 10^4$ and $< 10^5$; $\geq 10^5$ and $< 10^6$; $\geq 10^6$) which were entered into the model as factors. Covid-19 case fatality was retrieved from the death registries of the local Public Health Departments. Collinear variables were excluded. The best model was built by entering the predictors in a stepwise approach following the criteria of the significant contribution to the fitting of the model in terms of Pseudo-R squared. Alternative models including either categorized cVL, continuous cVL or raw Ct values were also built for comparison purposes.

To that end, the entire cohort was randomly divided into two sets (50% randomly selected samples): a model set (n = 3095), used to build predictive logistic regression models which were built using the fewest predictors achieving the best Pseudo R-squares, and an evaluation set (n = 3113) used to evaluate models' accuracy. The coefficients and constants of the models were included in predicting equations which calculated the probability of death (S1 File). The diagnostic performances of the models selected were evaluated using receiver operating characteristic (ROC) curve analysis, and optimal thresholds were obtained using the Youden J parameter. The predicted outcomes of the different models were utilised to calculate the respective sensitivity, specificity, positive and negative likelihood ratios, positive and negative predictive values, and overall accuracy. The latter statistical analyses were carried out using Analyse-it software (Analyse-it Software, Ltd, UK).

The cVLs from the end of February 2020 until October 2020 were reported using descriptive statistics.

For statistical purposes, the samples positive only at a target different from the N gene were considered positive with 0 copies/µL.

## Results

The original Ct values were compared with the Cts of the retested values to exclude the possibility of a degradation of the samples. No evidence of degradation was observed since the Cts were not statistically different between the retested values and the original test values (p = 0.74) (Fig 1). However, in terms of absolute value, a mean difference in the Ct of 1.6 and 1.9 was observed between all the samples (regardless of the primary assay) and only the Seegene samples (comparison restricted to samples initially assayed with the Seegene assay), respectively. The difference was not statistically significant (p = 0.45). This finding was not dependent on the original test used (Fig 1).

The dPCR assay performed adequately under the conditions described herein, achieving an LOD of 1.19 copies/µL (Fig 2).

Using a serial dilution experiment, the dPCR linearity was restricted to samples below 2.3 x$10^4$ copies/µL. Hence, in building the linear regression model, an adequate dynamic range was obtained by diluting those samples below the 22 Ct threshold 1:10 (Fig 3).

Finally, precision as a measure of inter-assay repeatability over the entire dynamic range achieved an average CV of 15.3% and a median CV of 4.3%: (S1 File).

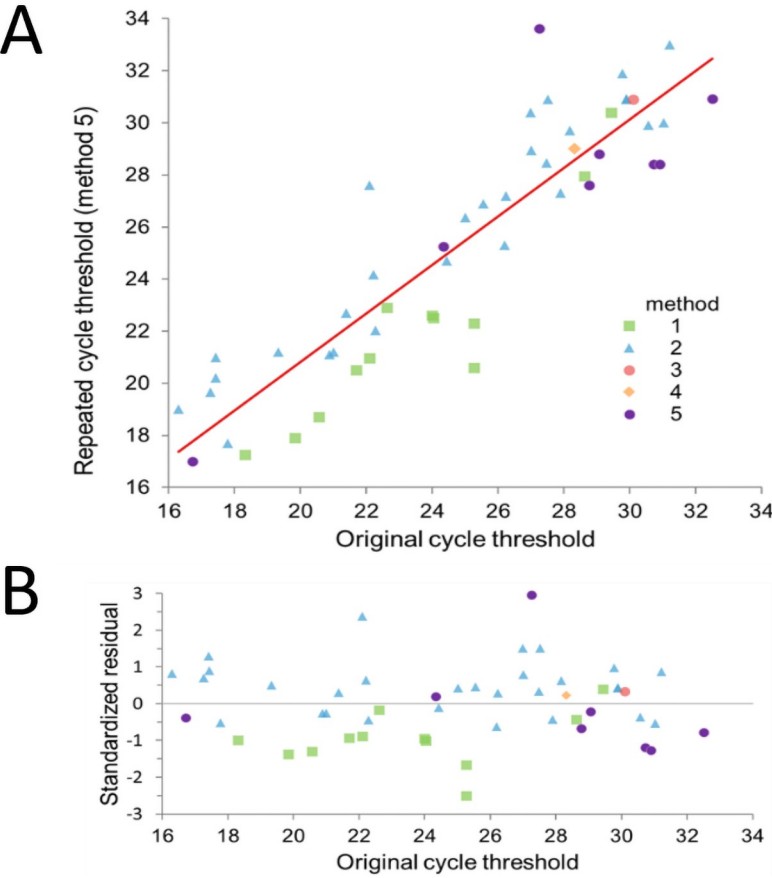

**Fig 1. Scatter plot A) and Residual plot B) of the original and repeated Ct values.** Colour codes indicate the original testing method.

The absolute quantification obtained by dPCR was regressed on the Ct values, and a linear regression function was derived. The model achieved good reliability and accuracy (Table 1).

The regression equation allowed calculating the absolute quantity of viral genome expressed as copies/μL in the evaluation set. Notably, in the evaluation set, the $R^2$ value was also 0.918 assessing a good linear correlation between the predicted and measured copies/μL values (Fig 4).

In absolute terms, the error in predicting the cVL expressed as MAD of the PE was 53.0%. The error was uniformly distributed into high, medium and low VL categories, although, to some extent, the latter showed higher errors. The complete comparison of measured versus calculated absolute copies/μL in evaluating the set counts are reported in detail in (S1 File).

The linear regression equation was used to calculate the cVL in a cohort of 6208 Covid-19 positive cases diagnosed in the period from 24 February to 30 September 2020 using the All-plex Seegene assay, and the cVL was recorded in the database. There is no unanimous consensus on how to interpret very low VL. In the present study, those cases with only one of the three positive target genes different from the N gene, which was that targeted by dPCR, was considered positive with 0 copies/μL [46]. The characteristics of the cohort are reported in Table 2.

The cVL differed greatly from 0 to more than $5x10^6$ copies/μL. The majority of cases showed less than 1 copy/μL (Fig 5).

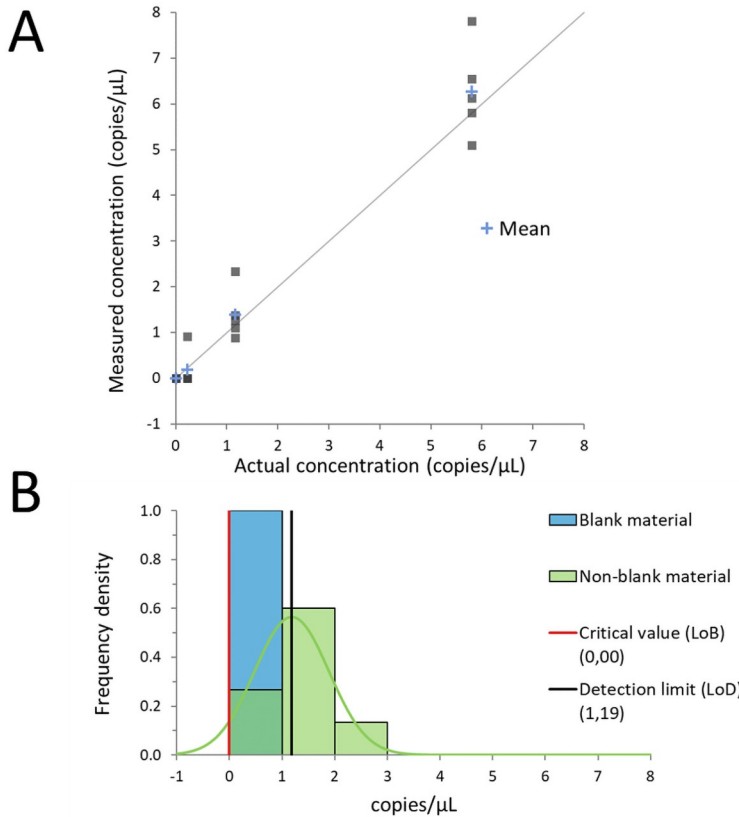

**Fig 2. Precision profile analysis of variance for assessing the analytical sensitivity of a digital PCR assay using α and β values of 5% for Limit of Blank (LoB) and Limit of Detection (LOD), respectively.**

By plotting the positive results over time, it could be observed that, in addition to the absolute number of positives, the cVL differed over time, being markedly lower during the summertime. The 90th and 95th percentiles of VL tended be very low during this period (Fig 5A). Furthermore, a rapid increase in VL in terms of higher percentiles, could be observed beginning in mid-August and peaking in September. This anticipated the exponential upwards rapid incidence increase of the epidemic curve observed in the same geographical area one month later (Fig 6B).

The cVLs were also examined after stratifying the cohort according to age group. Higher VLs, i.e. those which account for the majority of the transmission risk roughly estimated at 1500 copies/ μL by converting the reported Ct beyond which it is not possible to infect cell cultures using diagnostic samples [26], were observed primarily in the elderly followed by the youngest age group (Fig 7).

Finally, logistic regression models were used to investigate the effect of the sets of predictors considered in this study, including cVL, categorised cVL and raw Ct to predict the case-fatality outcome and to evaluate their accuracy.

Of all the possible models considered, the best one reached an adjusted R-square of 0.34 ($p < 0.01$) and included categorised cVL, age and gender. A high cVL was associated with increased case-fatality odds. In particular, a cVL $> 10^3$ copies/μL was significantly associated with increased mortality rates and a cVL $> 1 \times 10^6$ was associated with an Odds-ratio of 9.24 (CI 2.36–36.26; $p < 0.001$) (Table 3). Age (odds-ratio 1.11; CI 1.10–1.12; $p < 0.001$), and male

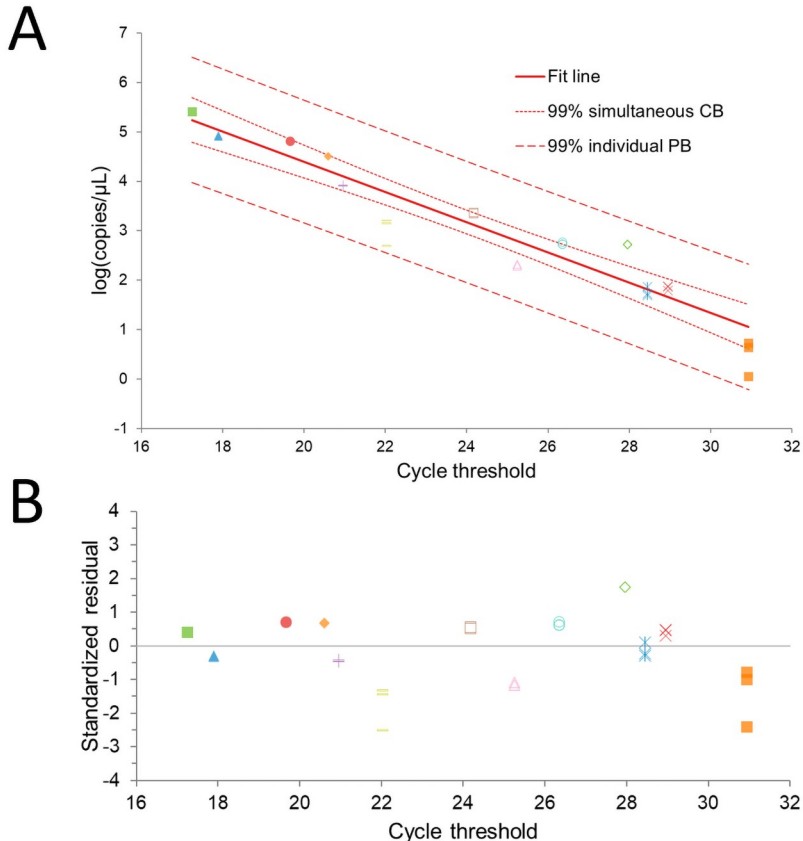

**Fig 3. Linearity fitting of the linear logistic model establishing the relationship between the cycle threshold values and the actual viral loads.** A) Linearity fitting plot including the individual plot of replicates (of the same colour) and the fitting line of linearity with the respective confidence bands and individual bands at the 99% level. B) Standardised residual plot: there are only two replicates of two different samples outside the 2 standard deviations.

gender (odds-ratio 1.51; CI 1.14–2.00; p<0.001) were also significantly associated with increased case-fatality odds.

The diagnostic performances of the models, including cVL, categorised cVL and raw Ct, were substantially equal having areas under the curve (AUCs) of 0.889, 0.888 and 0.889, respectively; no statistically significant differences were found at pair comparisons. At the optimal threshold, all models achieved very high specificity and low sensitivity. Being substantially equivalent, additional analyses were carried out using the model, including the cVL parameter. The optimal threshold was found to be 57.1% of the probability of death. Using this setting, the sensitivity, specificity, positive likelihood ratio, negative likelihood ratio, positive predictive value and negative predictive value were 21.7%, 99.0%, 5.70, 0.43, 67.0% and 93.0%, respectively (Fig 8).

In the evaluation set with a prior probability of case-fatality of 8.74%, the model identified 59 deaths out of 272. The false positive death predictions were 29 with a positive predictive probability of 67%. By fixing the sensitivity threshold at 75.0%, the predictive threshold was found at 2.66%. Using this threshold, a specificity of 83.7% was achieved and the model identified 204/272 case-fatalities; however, 464 false positive predictions occurred with a positive predictive power of 31% (Fig 9). Complete findings of the logistic predictive models are reported in S1 File. Both models performed almost identically regarding predictions (Fig 9).

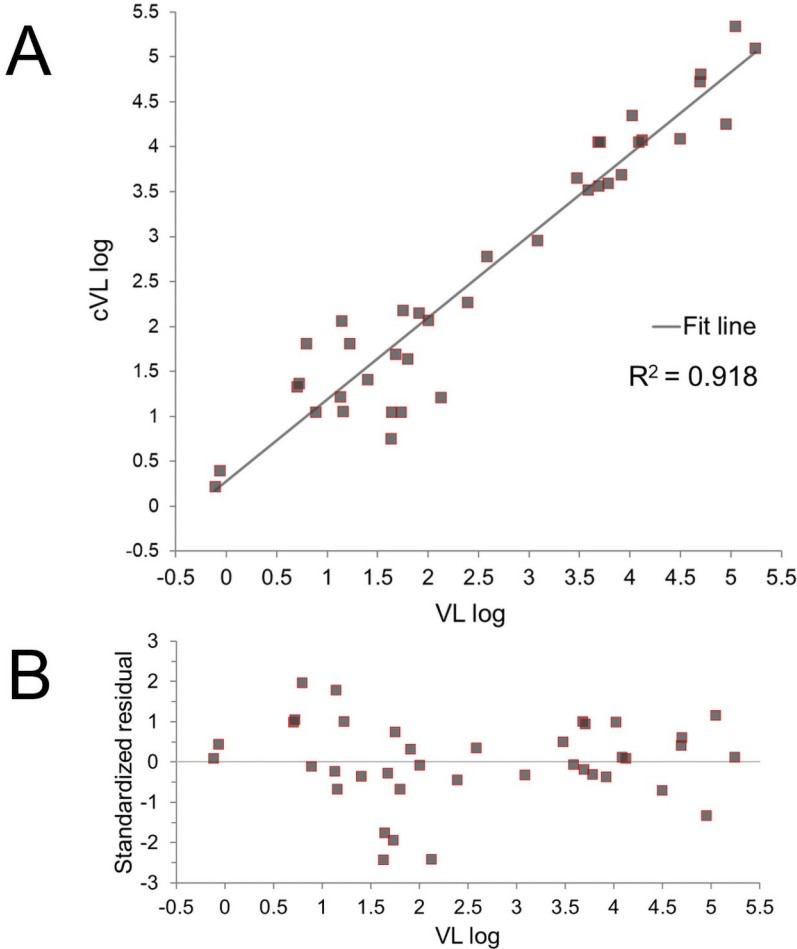

**Fig 4. A) Scatter plot and B) Residual plot of the findings achieved by the linear regression model in the evaluation set.** VL log: Log of the Viral Load value expressed as copies/µL; cVL log: Log of the calculated Viral Load.

## Discussion

In this study, a method of quantifying the VL by modelling the relationship between the dPCR and RT-qPCR results was established. To the best of the Authors' knowledge, this is the first study of this kind, although, while this paper was under review, another study was published suggesting the same approach for underpinning the investigations regarding Sars-CoV-2 biology [19]. The Authors further extended the approach by challenging the regression model in an evaluation set of data. The median of difference between the calculated and the measured VL was 53.0%. Since a 100% efficient PCR doubles the target every cycle, the 53.0% mean average error corresponds to a Ct value of less than 0.5. As the cVL spanned over 6 orders of magnitude from 0 to more than 5 million genomes/µL, the reported error could be considered almost negligible. Therefore, the model was used to calculate the VL in a cohort of 6208 cases diagnosed with Covid-19 having a known specified error.

Many studies have investigated VL in Covid-19 patients. Regrettably, the majority of them used an RT-qPCR assay originally intended as a qualitative, not a quantitative, assay; as a matter of fact, the Ct values of the diagnostic RT-qPCR in large cohorts of cases were used either as a rough estimate of the amount of virus as such or, in a minority of cases, were converted

Table 2. Characteristics of the cohort of SARS-CoV-2 positive cases (n = 6208).

| | |
|---|---|
| Age (median (IQR) | 55 (38–74) |
| Age categories (years) | n (%) |
| < 6 | 55 (0.9) |
| ≥ 6 < 18 | 298 (4.8) |
| ≥ 18 < 30 | 728 (11.7) |
| ≥ 30 < 50 | 1478 (23.8) |
| ≥ 50 < 70 | 1755 (28.3) |
| ≥ 70 | 1894 (30.5) |
| Gender | N (%) |
| female | 3155 (50.8) |
| male | 3053 (49.2) |
| Viral load (copies/µL) | n (%) |
| <1 | 2722 (43.8) |
| $\geq 1 < 10^1$ | 957 (15.4) |
| $\geq 10^1 < 10^2$ | 719 (11.6) |
| $\geq 10^2 < 10^3$ | 619 (10.0) |
| $\geq 10^3 < 10^4$ | 522 (8.4) |
| $\geq 10^4 < 10^5$ | 430 (6.9) |
| $\geq 10^5 < 10^6$ | 197 (3.2) |
| $\geq 10^6$ | 42 (0.7) |
| Turnaround time (days) | N (%) |
| 0 | 922 (14.1) |
| 1 | 4535 (73.9) |
| 2 | 708 (11.4) |
| ≥ 3 | 43 (0.7) |
| Swabs | n (%) |
| nasopharyngeal | 5982 (96.4) |
| oropharyngeal | 226 (3.6) |
| Ward/unit | |
| Hospital ward | 1289 (20.8%) |
| Emergency ward | 755 (12.2%) |
| Covid Drive-through | 890 (14.3%) |
| Preventive medicine unit | 3101 (50.0%) |
| Intensive care unit | 161 (2.6%) |
| Others | 12 (0.2%) |
| Presence of signs/symptoms | n (%) |
| no | 2315 (37.3) |
| yes | 2210 (35.6) |
| Not known | 1670 (26.9) |
| Outcome | n (%) |
| Deceased | 583 (9.4) |

IQR: Interquartile range.

into $\log_{10}$ copies/mL using an RT-qPCR calibration curve carried out once [47, 48]. However, in these studies, no detailed methods of Ct conversion were reported nor were the measured errors provided. A plethora of factors can affect the accuracy of absolute quantification by RT-qPCR using calibration curves. These concerns have recently been addressed together with

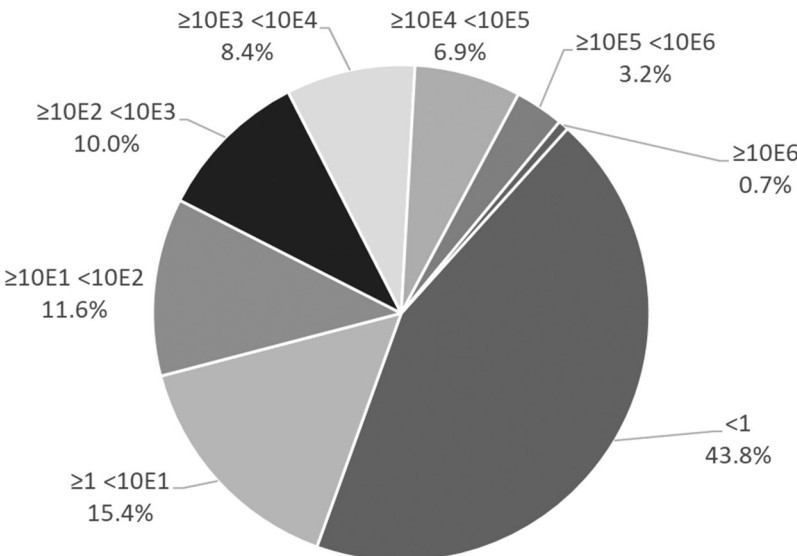

**Fig 5. Pie chart reporting the percentage of each viral load category in the cohort of 6208 Covid-19 cases.**
Calculated viral loads are expressed as $Log_{10}$copies/µL.

strategies for improving absolute quantification [49, 50]. One of the greatest concerns is that the efficiency of RT-qPCR may vary, biasing the accuracy and, specifically, the reproducibility of the calibration curves which, in turn, propagates the error. For the above-mentioned reasons, a direct comparison with the study herein reported could not be made, and the findings of these studies should be regarded as an error prone approximate estimation of the VL. Although hampered by the above-mentioned drawbacks, such studies established or not an association between VL and case-fatality; however, even in the former case, they did not provide a model for precisely quantifying the risk [4, 48, 51–54]. On the other hand, many fewer studies have quantified VL as copies/volume (of reaction or of sample) using dPCR. Remarkably, as indirect evidence of robustness and reliability of the calculated approach method, the measured copies/reaction in convalescent patients would be very similar, if calculated using the model here described from the high Ct values obtained in RT-qPCR [23]. Overall, the cVL range inferred with the regression model matches that directly measured from nasopharyngeal swabs in other studies [32]. However, to the best of the Authors' knowledge, no studies have quantified VL using diagnostic swabs and correlated it to outcome. Many advantages of the mathematical approach herein described are highlighted as 1) findings which can be compared between studies, 2) findings which can be included in a metanalysis or 3) different diagnostic RT-qPCR results which can be expressed in terms of Ct within the same laboratory and can be included in the same dataset which allows increasing the possibility of addressing the question of Sars-CoV-2 biology [19]. This latter advantage it is noteworthy since it has the potential to allow comparing the VL obtained using different RT-qPCR platforms in the same laboratory or even in different laboratories worldwide, provided that the linear regression equation was defined using the respective Ct data.

Overall, nearly half of the cases had less than 1 copy/µL. This quantity is very close to the detection limit of RT-qPCR. It is beyond the Authors' aims to investigate whether these were false positives in technical terms or true positive, carrying however only free nucleic acid and no viral particles. Moreover, the majority of studies which used viral isolation in cell cultures to estimate the infectivity identified in 24 Ct, the threshold beyond which the likelihood of

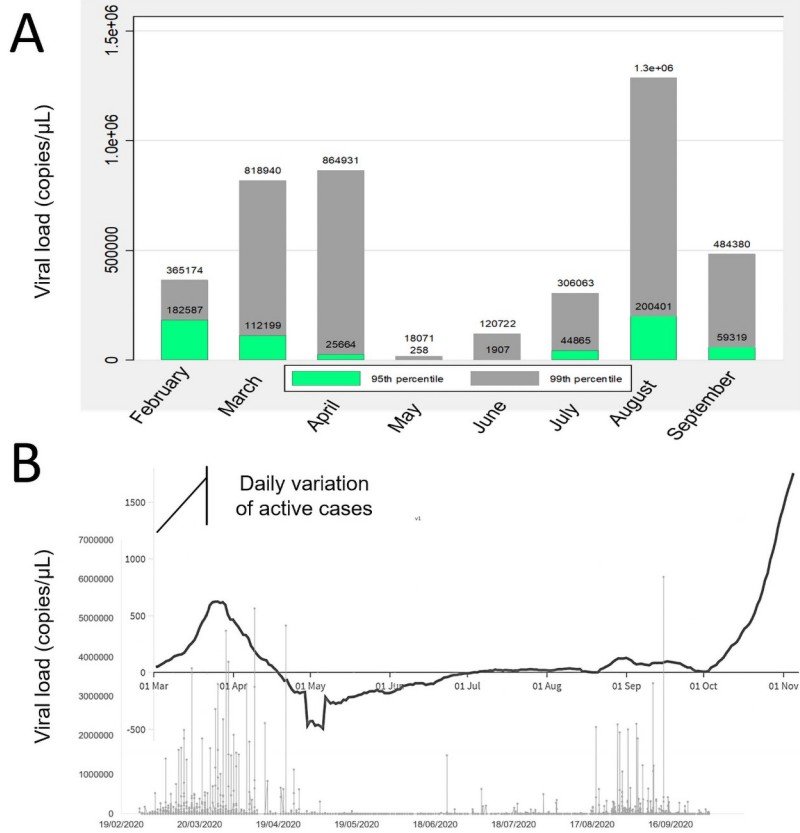

**Fig 6. Graph reporting the viral loads (VLs) over time (March 2020 to October 2020) in Italy.** A) histograms of the 90[th] and 95[th] percentiles of calculated VL on a monthly basis. B) Scatter plots of calculated VL (pale grey dots and lines) over the same timescale of fluctuations of active cases (black solid line) expressed as the last daily change (difference with respect to the day before) in active cases (Italian National Ministry of Health).

isolating SARS-CoV-2 from nasopharyngeal swabs drops abruptly, and in 30–33 Ct beyond which it is not possible to isolate the virus [26, 55, 56]. In terms of cVL according with our regression model, it could be roughly estimated that approximately 1500copies/μL represented the limit beyond which infectivity drops and 20 copies/μL the limit beyond which it is almost impossible to isolate the virus. These data are in agreement with those reported in a small case series including the absolute quantification of VL using RT-qPCR [57]. This study established a limit of $10^6$ copies/mL (corresponding to $10^3$/μL in the present study) for successfully achieving virus isolation. In the present cohort, fewer than 18% (17.4%) of the samples had a cVL > 1500 copies/μL and only 37.2% had a cVL > 20 copies/μL (Fig 5). The VL may also depend on the different times of diagnosis. Unfortunately, this represents an inherent limitation of the present study due to its retrospective nature.

With regard to the fluctuation of the VL, Clementi et al. (2020) reported low VLs during the summer period in Italy after the first epidemic wave had hit the country in the previous spring. Lower VLs were associated with fewer Covid-19 cases. The present study confirmed and extended these observations. After the first public health measures were eased in mid-May, the incidence continued to decline, reaching its lowest rate at the end of July 2020; similarly, the active cases also remained at very low levels until the end of September 2020 when an exponential rise in active cases was observed. Interestingly, this study confirmed that almost all

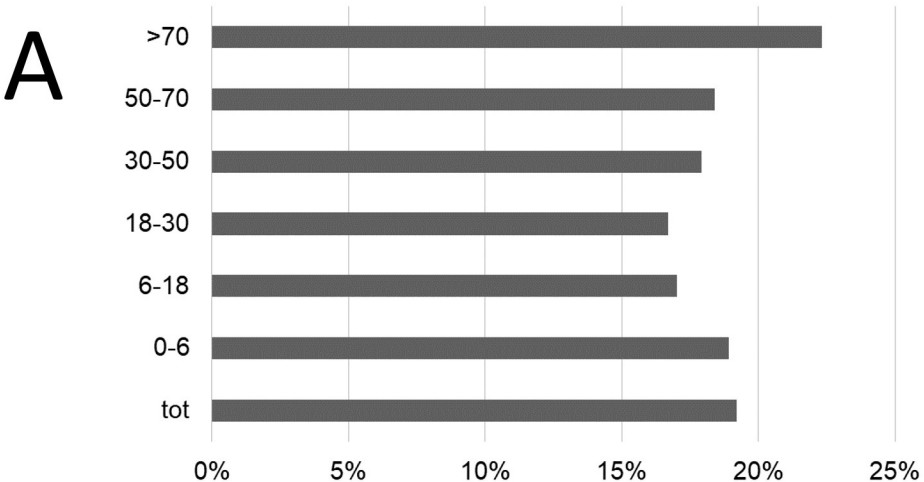

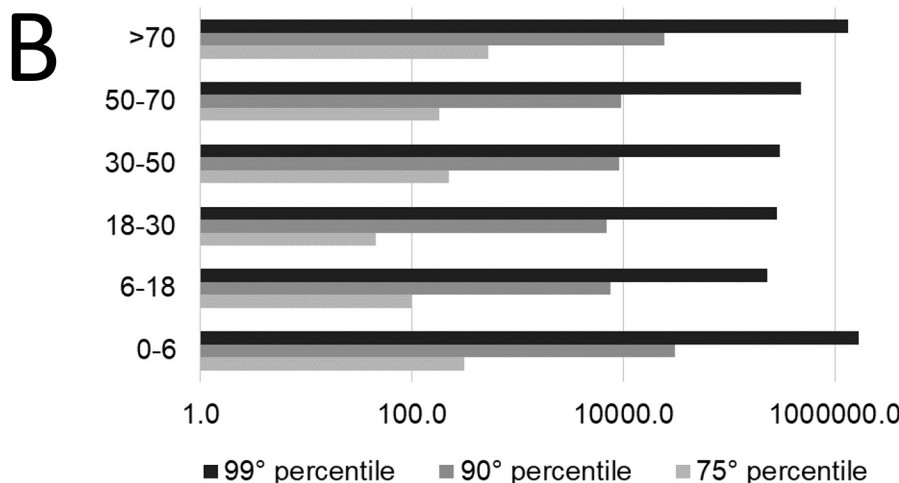

**Fig 7. Histograms characterising infectivity A) percentage of cases above the likely infectivity threshold (calculated viral load of 1500 copies/μL) divided by age category and B) 75th, 90th and 95th percentile values of viral load divided by age category.**

the cases diagnosed during the summertime had very low VLs. This evidence was corroborated by the 95th and 99th percentiles of VL data which were below the likely threshold of infectivity in May and June, and started to moderately rise in July and peaked in August. The peak of cases with high VLs was followed one month later by the exponential rise in incidence and active cases. This finding would suggest that an increase in VL should be considered as an early predictor of worsening epidemiologic parameters useful for tightening public health measures while minimising the economic impact of the restrictions [58–62].

The VL data were also applied to age groups to more specifically investigate the role of childhood in SARS-CoV-2 transmission. Although children are relatively spared by the severe forms of Covid-19, their possible role in transmission should be considered when

**Table 3. A three parameter (age, gender and cVL) logistic regression model to predict case-fatality.** The statistically significant (P<0.01) parameters are evidenced in **bold**.

Number of observations = 3095

LR chi$^2$ = 698.84

Probability > chi2 = 0.0000

Pseudo R2 = 0.346

| | Odds Ratio | SE | p | 95% Confidence Interval | |
|---|---|---|---|---|---|
| Viral load (copies/μL) | | | | | |
| <1 | | | | | |
| $\geq 1 < 10^1$ | 1.10 | .23 | .962 | .65 | 1.58 |
| $\geq 10^1 < 10^2$ | 1.30 | .30 | .261 | .82 | 2.05 |
| $\geq 10^2 < 10^3$ | **2.30** | .52 | **.000** | 1.47 | 3.58 |
| $\geq 10^3 < 10^4$ | **2.63** | .65 | **.000** | 1.62 | 4.27 |
| $\geq 10^4 < 10^5$ | **2.23** | .57 | **.002** | 1.35 | 3.69 |
| $\geq 10^5 < 10^6$ | **4.70** | 1.53 | **.000** | 2.49 | 8.89 |
| $\geq 10^6$ | **9.24** | 6.45 | **.001** | 2.36 | 36.26 |
| Age | **1.11** | .007 | .000 | 1.10 | 1.12 |
| Male gender | **1.51** | .219 | **.005** | 1.14 | 2.00 |

SE = Standard Error; LR = logistic regression.

implementing radical measures, such as school closure [3]. The authors found that, in addition to the elderly, preschool children (0–6-years old) had the highest VL in both the higher percentiles and in the percentage of cases above the aforementioned threshold of infectivity (Fig 7). Conversely, school children >6-years old and < 18-years old) were those with the lowest VLs. To the best of the Authors' knowledge, the largest cohort study to date has showed a moderate trend to higher VLs with increasing age categories [63, 64]. However, it should be emphasised that, in this study, the percentiles above the threshold considered as the limit of infectivity were almost similar among all age categories. This latter parameter is likely more representative of the weight of each age category as a transmitter.

Strong evidence exists that the absolute quantification of circulating VL is an independent and strong predictor of fatality [34–38]. However, this approach requires invasive blood sampling which is carried out solely in hospital settings with prognostic aims while its diagnostic value is limited. Herein, the power of cVL, and some other simple and readily available signalment data at the moment of Covid-19 testing were evaluated. The regression model found cVL to be an independent predictor of case-fatality after correcting for gender and age. In particular, a VL above the threshold of $10^6$ copies/μL was strongly associated with negative outcomes. Similar findings have also been reported by others [48]; however, in the present study, the odds ratios were additionally refined using different levels of cVL, making the VL readily interpretable. The other negative independent predictors found were male gender and older age. These predictors were almost invariably found in all the studies and meta-analyses carried out in hospital settings [48, 65–67], with or without considering VLs. When the model including the three predictors was used to predict the outcome in the evaluation set of cases, it was notably able to specifically detect those cases having a high probability of survival. For instance, using the optimal threshold, the model identified 3024 out of 3113 subjects who were predicted to survive the Sars-CoV-3 infection with a probability of 93% (negative predictive value). Of the 89 predicted deaths, eventually 59 died; hence, the model showed a 66.3% positive predictive value. Conversely, if adequate sensitivity was requested by the model, for instance 75%

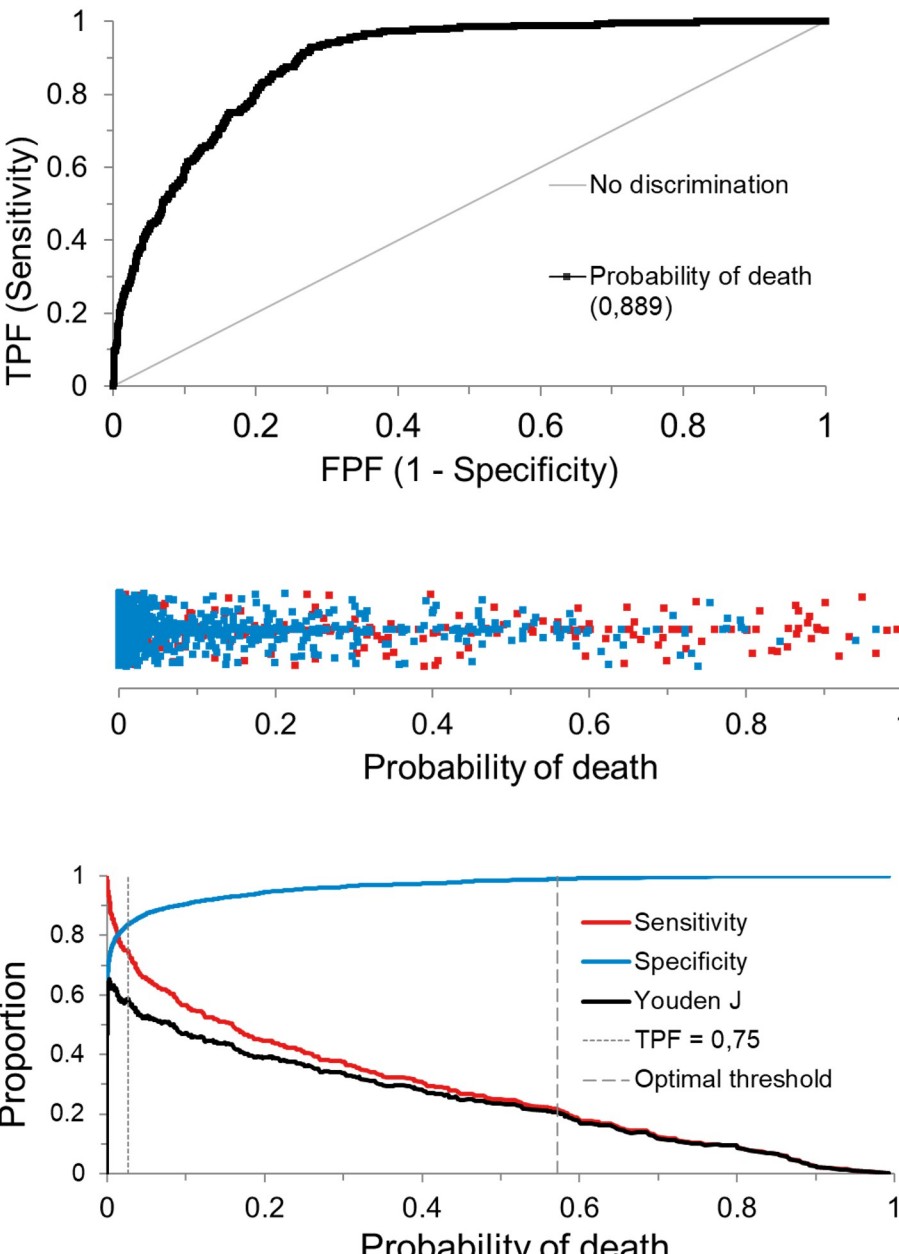

**Fig 8. Diagnostic performance of the model including the predictors age, gender and calculated viral load.** A) Receiver operating characteristic curve of the model showing an area under the curve of 0.889. B) Scatter plot showing the survivors (blue) and the deceased (red) along with the predicted probability of death calculated by the model. C) the Youden plot highlighting the optimal threshold and the threshold at a fixed sensitivity value of 75%. TPF: True positive fractions (sensitivity). FPF: False positive fractions.

sensitivity, the model allowed identifying 204 out of 272 case-fatalities although 465 false positive cases were also found.

This evidence may have relevant implications in terms of public health since this tool could give public health institutions the opportunity of classifying those patients at risk of death already at the moment of diagnosis so as to efficiently allocate finite health resources by

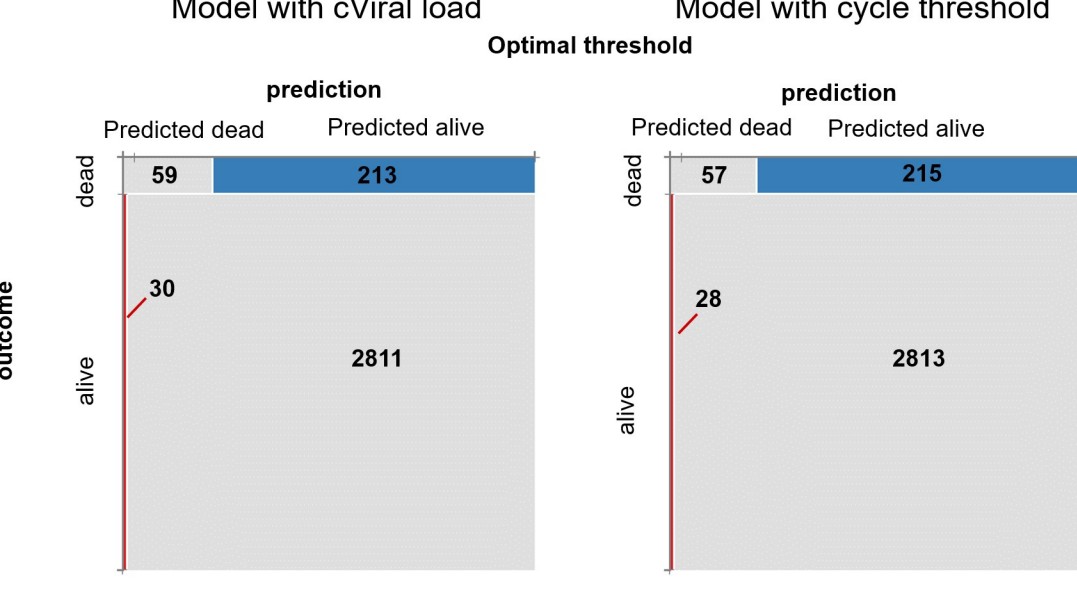

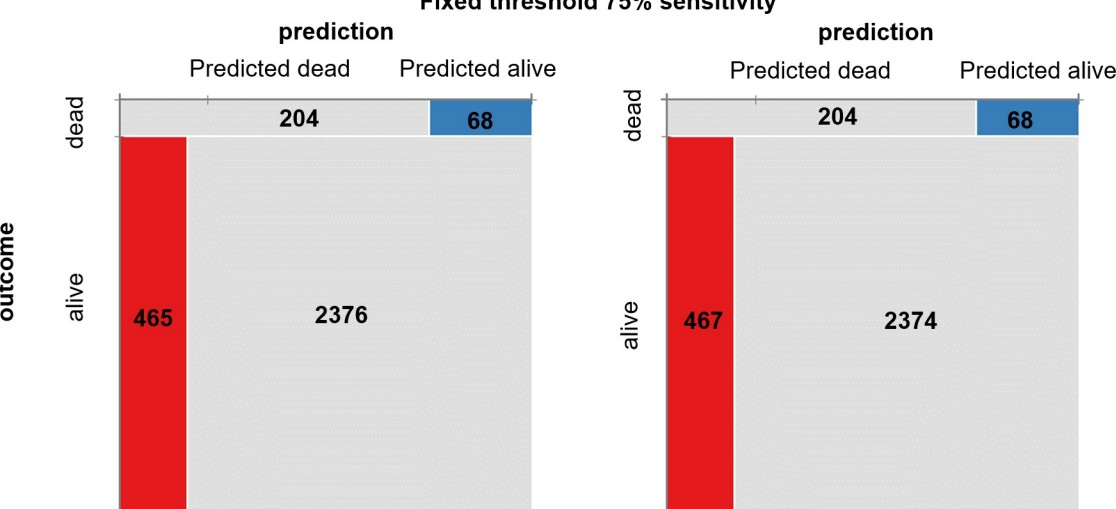

**Fig 9. Mosaic plot of models including either calculated viral load or raw cycle threshold to classify patients at risk of death.**
Both models performed almost identically. A) Using optimal thresholds, both models showed very high specificity but low sensitivity. Conversely B) at the fixed threshold of 75%, of sensitivity the specificity dropped to approximately 84%. Within the mosaic boxes, the number of subjects from the 3113 of the evaluation set are indicated.

focusing medical monitoring on those high-risk patients at an early stage, at a very low additional cost. Furthermore, the threshold may be set at different levels based upon specific aims and guidelines in order to focus the public health resources on those Covid-19 cases at risk of developing severe disease.

There are many reviews and meta-analyses which have investigated the risk factors associated with death outcome. However, the majority of them examined different cohorts of patients variably selected, i.e., those hospitalised, those having the presence of specific comorbidities, those coming from specific wards and those presenting specific markers. All these

studies were suitable for stratifying patients within specific settings, implementing in an evidence-based manner, the resource allocations. It should be noted that this was the first study aimed at addressing the role of VL as an independent variable in a cohort of diagnosed cases from the Diagnostic Laboratory. This evidence could be important for the early stratification of cases, thus focusing medical surveillance on patients at a high risk of developing severe forms of Covid-19 and on efficiently allocating resources. Interestingly, this result could be attained computationally by simply including the signalment and anamnestic data already available at the moment of diagnosis along with the cVL data. Unfortunately, data regarding the presence or absence of signs and symptoms were very incompletely represented in the database; therefore, they could not be included in the model. It is very likely that this information would have allowed better stratifying the sample. Public health laboratories should be aware of this and improve the exchange of information between all the players engaged in the network of diagnosing and curing Covid-19.

## Supporting information

**S1 File. Supplementary materials & methods.**
(DOCX)

**S2 File. Complete dataset.** The Excel file includes two spreadsheets providing both model group and test group. In the latter, the predictive formulas are embedded in the spreadsheet.
(XLSX)

## Author Contributions

**Conceptualization:** Fabio Gentilini, Maria Elena Turba, Vittorio Sambri.

**Data curation:** Francesca Taddei, Tommaso Gritti, Michela Fantini, Giorgio Dirani.

**Funding acquisition:** Maria Elena Turba, Vittorio Sambri.

**Investigation:** Fabio Gentilini, Maria Elena Turba, Francesca Taddei, Tommaso Gritti, Michela Fantini, Giorgio Dirani, Vittorio Sambri.

**Methodology:** Fabio Gentilini, Maria Elena Turba, Francesca Taddei, Tommaso Gritti, Michela Fantini, Giorgio Dirani, Vittorio Sambri.

**Resources:** Maria Elena Turba, Vittorio Sambri.

**Supervision:** Fabio Gentilini, Vittorio Sambri.

**Validation:** Fabio Gentilini, Michela Fantini, Giorgio Dirani.

**Writing – original draft:** Fabio Gentilini, Maria Elena Turba.

**Writing – review & editing:** Vittorio Sambri.

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
