## [Decision Letter · Decision Letter 0]

14 Jul 2021

PONE-D-21-12462

Modelling RT-qPCR cycle-threshold using digital PCR data for implementing SARS-CoV-2 viral load studies

PLOS ONE

Dear Dr. Gentilini,

Thank you for submitting your manuscript to PLOS ONE. After careful consideration, we feel that it has merit but does not fully meet PLOS ONE’s publication criteria as it currently stands. Therefore, we invite you to submit a properly revised version of the manuscript that addresses the points raised during the review process.

Notably and also according to the comments of the expert reviewer

#1 Most importantly a careful and appropriate statistical analysis

#2 Appropriate quantitative RT-PCR analyses serving as a standard curve based on effective copy numbers of the viral RNA. Based on such a standard real VL (viral loads) numbers can be provided.

#3. there are some doubts on the consistency and reproducibility of the data; this is of the utmost importance , see #5 of the reviewer. Yet this section is in need of clarification.

We look forward to receiving your revised manuscript.

Kind regards,

Jean-Luc EPH Darlix, MG, Ph.D.

Academic Editor

PLOS ONE

Journal Requirements:

2. Please list all of the different RT-qPCR assays used in the experiment.

3. Please clarify if the biological samples used in your study were:

(1) from an established biobank (if so please provide the name and a link)

(2) specifically collected for this study or not

(3) collected through a medically prescribed test

(4) completely de-identified before researchers accessed the samples

4. In your ethics statement in the Methods section and in the online submission form, please provide additional information about the cohort used in your study. Specifically, please ensure that you have discussed whether all data were fully anonymized before you accessed them and/or whether the IRB or ethics committee waived the requirement for informed consent. If patients provided informed written consent to have data from their medical records used in research, please include this information.

Reviewers' comments:

Reviewer's Responses to Questions

**Comments to the Author**

1. Is the manuscript technically sound, and do the data support the conclusions?

Reviewer #1: Partly

2. Has the statistical analysis been performed appropriately and rigorously? 

Reviewer #1: No

3. Have the authors made all data underlying the findings in their manuscript fully available?

Reviewer #1: No

4. Is the manuscript presented in an intelligible fashion and written in standard English?

Reviewer #1: Yes

5. Review Comments to the Author

Reviewer #1: Gentilini et al present a manuscript describing a method to estimate the viral load of SARS-CoV-2 based on the Ct values of RT-qPCR. They performed digital PCR for a small number of samples to build a linear regression model to infer the viral load (copies per µl), then applied it to a much larger data set. Inferred viral load over time revealed that it preceded the peak of positive cases. Furthermore, logistic regression using inferred viral load and sex and age was able to predict mortality at an above chance level. Although this manuscript is potentially useful, the authors need to provide some more validation and head-to-head comparisons between Ct and calculated viral load (cVL) to clarify the advantage of their method.

Major comments:

1. Instead of using COVID-19 patient samples, the authors first need to show the results of both RT-qPCR and dPCR by using a concentration series of SARS-CoV-2 RNA standards that have known copy number per µl. It is important to show the sensitivity and dynamic range of these methods. This experiment also helps to properly infer the actual copy number based on dPCR results, especially since the Ct values varied across retesting. Currently, it does not appear that this crucial control experiment was performed.

2. To clearly show this calculated VL value is more useful than the Ct value, the authors need to do the same analyses and show if their methods indeed can provide more insight in terms of the relationship between test results and COVID-19 mortality or active case dynamics. For example, they need to show how better is the performance of calculated-VL based logistic regression compared to Ct-based model. If, as the authors claim, the Ct values and VL are linearly related, then one may not expect any improvement between these new metrics.

3. The performance of the logistic regression is not convincing enough as a useful tool. Given the low rate of mortality (~9%), ~90% accuracy of their logistic regression is not particularly high (you can achieve >90% accuracy even if you called everything negative). In a model with 50% threshold, sensitivity is ~25%, which means only a quarter of the true positives were correctly called as positive. In addition, positive predictive value is 63%, which means only 2/3 of the positives this model called are true positive. Even though the authors discussed the potential use of this prediction as a tool to screen more susceptible patients, it is hard to imagine this model can actually be used for that purpose.

4. The authors purified RNA from UTM stored samples and compared results of RT-qPCR and dPCR. It has been shown that RT-qPCR results are different between purified RNA and UTM samples as shown in Figure 3 of reference 31, https://www.nature.com/articles/s41598-020-80715-1). The authors need to describe how original tests these 51 samples and other ~6000 samples were performed in more detail (such as if purified RNA was used for RT-qPCR). Although it seems a part of 51 samples and all of ~6000 samples were tested by “Seegene assay”, the authors did not describe the detail of this method. If these original tests were performed directly from UTM samples, using purified RNA for building a regression model does not sound reasonable.

5. I am not sure what is the rational of performing dPCR only once as described in line 161. The authors should perform multiple times to see the consistency. Related to this, it appears that the authors performed the regression with a single training and test split, but it would be better to perform k-fold cross validation with multiple restarts (meaning randomly separate all samples to training and test sets many times). This would help to understand if their model is generalizable or specific to the particular training/test (evaluation) sets. Additionally, the percentage of data in the training and evaluation sets differs between the linear and logistic regression models.

6. The authors need to show more data instead of just reporting the results of statistical tests or analyses by numbers or table format. This manuscript is not reader-friendly partly due to the lack of plots for most of their key analyses. Individual points will be pointed out below. The better visualization is also important to evaluate their key results such as the performance of linear and logistic regression.

7. The authors claimed there was no significant difference between retested and original Ct values for 51 samples they analyzed. They need to show a plot showing the relationship between retested and original values for both SARS-CoV-2 genes and control genes. It is important to show how consistent these values are to evaluate their results.

8. The authors need to show plots showing Ct values and log10(measured VLs) with a regression line for 51 samples they analyzed in addition to table 2. This will help readers to evaluate their model. Based on the table, the difference between measured and calculated VLs is not small even though they their regression was significant, and it is hard to understand the pattern of distribution without a plot. Additionally, the R-squared value (0.900) on the training set is quite low, especially for data that is distributed across multiple log-orders. Likewise, the MAD on the evaluation data is >50%, with some samples being off by up to 10-fold, suggesting that the model does not generalize well.

9. The results of logistic regression should be also presented by some plots instead of just showing numbers as a table. The methods also describe using the VL as both a continuous and categorical variable, but only the results from the categorized version are shown in table 4. The authors need to show both results. It would also be helpful to show key values such as sensitivity as a function of probability threshold. Even though the authors used three thresholds, using more thresholds and showing them as plots would be more helpful.

Minor comments:

1. It would be more helpful for readers to generate a figure explaining the experimental layout, which is nicely described by text in the Materials and methods section.

2. In page 10 (line 181), the authors cited “Supp. Mat” though I was not able to find corresponding description or data in the attached supplementary materials.

3. Figure 1 needs a label saying “calculated viral load (copies/µl)”

4. Figure 2 was not properly labeled. The authors need to show the axis labels and what black curve and grey histogram indicate. Although the text mention about 90th and 95th percentile by citing this figure, it doesn’t look those values are properly presented in this figure. I think it looks a lot better to have 2 stacked plots instead of current partially overlaid one.

5. Infectivity threshold at 1500 copies per µl sounds somewhat arbitrary. The authors need to cite references or explain more.

6. Although the authors cite Vasudevan et al. 2021, they should also include more discussion about how their results compare to previous efforts to use digital PCR to quantify VL.

7. The authors should explain and justify their sample sizes for their linear and logistic regression models. The original linear regression model is fit on only 13 samples, but then applied to over 3000 samples in the logistic regression model.

8. The authors should add original and retested Ct values in table 2.

9. The authors should provide a supplementary table that include all the information about patients (age, sex, etc) and viral loads (Ct, inferred VL) for ~6000 samples they used.

6. PLOS authors have the option to publish the peer review history of their article (what does this mean?). If published, this will include your full peer review and any attached files.

Reviewer #1: No

---

## [Author Response · Author response to Decision Letter 0]

2 Oct 2021

Editorial Office of

Plos One

Ozzano dell’Emilia, September 24, 2021

Modelling RT-qPCR cycle-threshold using digital PCR data for implementing SARS-CoV-2 viral load studies

Dear Editor, 

We would like to re-submit the revised manuscript (PONE-D-21-12462)) for publication in your journal. The manuscript has been completely revised as suggested by the reviewer. To address his/her concern great effort has been made to re-analyze the raw dataset by carrying out new analyses and generating new plots. As a matter of fact, we could claim that the concerns and criticisms have been wholly addressed although, the reviewer showed overall skepticism on the authors’ approach although he/she acknowledged some merits of the paper. We would respectfully draw the attention of the Editorial Office, and of the reviewer as well, to the paper published very recently, in particular after the submission of the present paper entitled “ SARS-CoV-2 RNA Quantification Using Droplet Digital RT-PCR”, The Journal of Molecular Diagnostics 2021 23,8, Pages 907-919 published on May 29, 2021. The manuscript describes the hypothesis of using mathematical modelling (linear regression) to model the relationship between the absolute count carried out using ddPCR and the cycle threshold. We quote ”.. RT-ddPCR derived SARS-CoV-2 E gene copy numbers were further calibrated against cycle threshold values from a commercial real-time RT-PCR diagnostic platform. This log-linear relationship can be used to mathematically derive SARS-CoV-2 RNA copy numbers from cycle threshold values, allowing the wealth of available diagnostic test data to be harnessed to address foundational questions in SARS-CoV-2 biology. Furthermore, no studies to our knowledge have calibrated SARS-CoV-2 viral loads to diagnostic test Ct values.“ Our manuscript was aimed at addressing some questions in SARS-CoV-2 biology using precisely that approach. 

We would like to thank the reviewer for giving us the possibility to provide in detail the diagnostic method. We think that we have fully addressed his/her concerns and have hopefully improved the quality of the manuscript. 

The revisions suggested are indicated in red in the revised manuscript.

Since the rebuttal is very long, we eventually decided to not submit it for review by a mother tongue person. We hope that our answers herein reported are clear enough for the reviewer. However, we regret any errors contained herein. 

Below detailed answers to the concerns raised:

Reviewer #1: Gentilini et al present a manuscript describing a method to estimate the viral load of SARS-CoV-2 based on the Ct values of RT-qPCR. They performed digital PCR for a small number of samples to build a linear regression model to infer the viral load (copies per µl), then applied it to a much larger data set. Inferred viral load over time revealed that it preceded the peak of positive cases. Furthermore, logistic regression using inferred viral load and sex and age was able to predict mortality at an above chance level. Although this manuscript is potentially useful, the authors need to provide some more validation and head-to-head comparisons between Ct and calculated viral load (cVL) to clarify the advantage of their method.

As suggested, we have carried out some more head-to-head comparisons and further statistical analysis and reported in more detail the validation process of dPCR. However, we would refer the reviewer to the above-mentioned paper by Kinloch et al., 2021 (just published) who also used the same mathematical modelling approach. 

Major comments:

1. Instead of using COVID-19 patient samples, the authors first need to show the results of both RT-qPCR and dPCR by using a concentration series of SARS-CoV-2 RNA standards that have known copy number per µl. It is important to show the sensitivity and dynamic range of these methods. This experiment also helps to properly infer the actual copy number based on dPCR results, especially since the Ct values varied across retesting. Currently, it does not appear that this crucial control experiment was performed.

This is a key point. Unfortunately, we only partially agree with the reviewer on this crucial point. Before we answer this issue, we should share the assumption that digital PCR has been widely demonstrated to be the most accurate way to assess the true value of a standard. Measuring the copies/µL in a “clinical sample” is better than synthetic nucleic acids and then titrated viruses. Hence, the use of clinical samples quantified with dPCR is at least equivalent (if not better) than other standards (i.e. synthetic nucleic acids, quantified by fluorimetric methods, or titrated viruses spiked in negative samples). That said, we actually carried out a validation study on the dPCR but, since it was not within the main aim of the study, the findings had been reported in brief in the original manuscript which the reviewer may have missed: quoting “Since the linearity of dPCR is limited by the fixed capacity of the vessels on the chip, it is extremely important to define the dynamic range of linearity. To that end, a dilution experiment was carried out; a cDNA sample which had been highly positive at qPCR with a Ct of approximately 16 was serially diluted 1:10 in molecular biology grade water and assayed in duplicate with dPCR. Since the first 2 dilutions were beyond the chip saturation, only the last five dilutions were linear, achieving a good r coefficient of 0.994. Based on these findings, all the samples >22 Ct were tested in dPCR as such while all the cDNA samples < 22 Ct were pre-diluted 1:10 in 5mM Tris-HCl. “it should also be taken into account that the same primers (CDC N gene) and method had already been validated in many studies (see references of the manuscript) with very consistent results. We did carry out linearity assays and analytical sensitivity experiments of the dPCR assay; the linearity experiment is relevant inasmuch as the dynamic range could affect the actual copies/µL of the standards (clinical samples). This issue is well known; hence we ascertained that a dilution was necessary for those samples < 22Ct. In the revised manuscript many more details are reported and graphed either in the main text or in Suppl. Materials. In detail, the analytical sensitivity has been assessed as LOD and reported in the revised manuscript adding some sentences in methods, results and a new Figure. We do not think that LOD is as critical as the dynamic range since the dPCR has been used here to create a linear regression model and not to diagnose the COVID-19. So, it is not evident to us the importance of including this info but we did. Demonstrating that the clinical samples used to regress Ct on the copies/µL and claiming that the clinical samples are valid and the copies/µL reliable, the reviewer suggested establishing the same features for the RT-pPCR assay. Again, we do not agree that this issue is relevant to the aim of the paper since our regression model includes and corrects the inherent bias of the Ct (referenced in the manuscript). However, such info could be obtained from another paper from our group. So, we eventually decided to cite the manuscript where the readers could draw data and graphics on this issue. (Brandolini M, Taddei F, Marino et al. (2021) Correlating qRT-PCR, dPCR and Viral Titration for the Identification and Quantification of SARS-CoV-2: A New Approach for Infection Management. Viruses. 28;13(6):1022.). It could be seen that for values > 27 Ct the repeatability of the qPCR worsens greatly. Overall, a R2=0.9128 was found. 

All these further findings have been cited in the revised manuscript to hopefully improve the quality of the manuscript. 

2. To clearly show this calculated VL value is more useful than the Ct value, the authors need to do the same analyses and show if their methods indeed can provide more insight in terms of the relationship between test results and COVID-19 mortality or active case dynamics. For example, they need to show how better is the performance of calculated-VL based logistic regression compared to Ct-based model. If, as the authors claim, the Ct values and VL are linearly related, then one may not expect any improvement between these new metrics.

In this regard, we have to re-affirm that in the original manuscript we did not claim that cVL would have performed better than the respective CT value since in this setting Cts are likely very homogeneous and hence our aim was not to compare raw Ct against cVL but instead to demonstrate the reliability of using the cVL. Using cVL has many advantages which were pointed outt in the revised manuscript. However, the comparison between Ct and cVL (and even he continuous and categorized one) was reported and graphed in the revised manuscript as requested. 

That said we have also to stress that 1) clearly the use of dPCR as a diagnostic method due to its superior accuracy than qPCR would have achieved more accurate Viral Load quantification. Unfortunately, this is not good due to costs and turn-around time issues. Regardless of whether Viral Loads are prognostic this is a matter of investigations as in our paper. We have demonstrated that the addition of cVL(or raw Ct) to age and gender increased significantly the predictive power. Conversely, as suggested by the reviewer the model with cVL is almost equivalent to Ct (cVL is slightly but not significantly better). This was quite expected for the reasons that the reviewer pointed out. Precisely because of this, we have not claimed in the manuscript that the model with cVL would be better than the model with Ct. Differently, changing settings (labs, methods, instruments, kits, etc.), Ct may vary greatly and the model would likely lose accuracy. So, the improvement is not between the Ct and cVL in the experimental setting but in the real world. If we talk about Viral Load why would it be better to use a crude semiquantitive estimate instead of a more standardized and comparable parameter? We would also stress that we have measured the error rate of our method. It makes no sense to measure the error rate of other methods inherently more prone to errors (Please, see references). 2) CT values obtained by qPCR is not a quantitative method but it is a rough estimate of viral load. We cannot compare Ct against copies/µL. Basically, Ct may be compared within the same environment, personnel, qPCR method and so on but they are not comparable outside these settings in such a way that the eventual prognostic power in a setting cannot be generalized 3) Also viral load measured using Ct obtained with qPCR and interpolated using a calibration curve is more prone to errors than dPCR for a multitude of reasons addressed in detail elsewhere (References in the text). Even worse is that many studies used viral load using a calibration curve obtained only once. This is even worse than using the Ct. 4) We measured the error rate. It is not soundly based to say that since others did not, now we have to measure the errors of others. You can intend our approach as we do not know if this quantitive method employing the cutting-edge, most accurate, most reproducible, most precise, most sensitive technology is better but at least we have measured the error of measurement. This would be enough to prefer this approach rather than the Ct approach without measuring the error. 

Also, the assertion that since two sets of data are linearly correlated, hence one or the other is the same is conceptually wrong from a mathematical standpoint. It’s possible but not necessarily true. Indeed, it depends on how the Ct values are dispersed around the regression line. The best way to explain this is by looking at the Residuals. Residuals may lay on the regression function (in this case the regressed value or the original datapoint is the same) or lay far from the regression line and in this case the use of the regressed data would be better than the original datapoint. In other words, errors are dispersed around the regression functions. Only those samples close to or overlapping with the regression line may give the same prediction as the data obtained by the regression line while those samples laying outside (for instance this is more evident for Ct values > 27, See the graph in Brandolini et al. 2021) the regression line and so the prediction based on the measured value is likely farther from the correct value than the predicted one. Unfortunately, low values even if incorrect are likely not predictive of case-fatality and hence the models using the raw Ct or cVL gave similar performances even if raw Cts are incorrect when > 27. So, it is not obvious that Ct and calculated Viral loads had the same predictive power. In our case, the Ct values are more prone to error for low Ct values which are, in turn, those values much less prognostic than higher values.

So, the same regression line may be obtained by different sets of data more close or less close to the line. Clearly, each set of data would have a different impact in predicting a variable even if they had the same regression line.

However, in the revised manuscript the methods, results and discussion sections have been revised and two novel Figures have been added. We hope they will be appreciated by the reviewer, especially the Figures.

3. The performance of the logistic regression is not convincing enough as a useful tool. Given the low rate of mortality (~9%), ~90% accuracy of their logistic regression is not particularly high (you can achieve >90% accuracy even if you called everything negative). In a model with 50% threshold, sensitivity is ~25%, which means only a quarter of the true positives were correctly called as positive. In addition, positive predictive value is 63%, which means only 2/3 of the positives this model called are true positive. Even though the authors discussed the potential use of this prediction as a tool to screen more susceptible patients, it is hard to imagine this model can actually be used for that purpose.

This is the reviewer’s opinion. Unfortunately, we disagree with this opinion which represents a very original approach to statistical modelling. How much is “enough” it is a matter of opinions. We have reported data. 90% accuracy with 9% mortality. It is indeed 90% accuracy. We would like to stress that 1) during the pandemic, we approved IVD assays (i.e. SARS-CoV-2 antigenic tests) that have been approved achieving no more than 70% accuracy. Also, molecular assay with a 98% accuracy that with disease prevalence of 1-5% means that tests do make errors either false positive or false negative. Also, predictive models may do errors. Our model has great benefits: first of all there is no cost. Indeed, it relies on already available data to obtain an objective prediction. Second it is straightforward. Third it allows concentrating the effort of public health services on a smaller subset of subjects at higher risk. If this is worthless for the reviewer, it is just an opinion and likely an opinion of a person not directly involved in the rational use of public resources in handle the pandemic. However, the example does not fit at all. The reviewer cannot take extreme cases and make a rule. In the real world of Public Health Services, among more than 3000 Covid-19 cases, at no cost the model predicted 111 deaths. Thus, the PHS may prioritize the assistance on 111 predicted death cases among >3000. Within these predicted deaths, 70 were correctly identified (They would actually die). This means that the PHS could prioritizes those patients who had 63% of possibility of death. Evidently, this critical impact of the study was not highlighted enough in the first version of the manuscript. As a consequence, a paragraph has been added to the revised manuscript on lines 399-404.

4. The authors purified RNA from UTM stored samples and compared results of RT-qPCR and dPCR. It has been shown that RT-qPCR results are different between purified RNA and UTM samples as shown in Figure 3 of reference 31, https://www.nature.com/articles/s41598-020-80715-1). The authors need to describe how original tests these 51 samples and other ~6000 samples were performed in more detail (such as if purified RNA was used for RT-qPCR). Although it seems a part of 51 samples and all of ~6000 samples were tested by “Seegene assay”, the authors did not describe the detail of this method. If these original tests were performed directly from UTM samples, using purified RNA for building a regression model does not sound reasonable.

The reviewer is totally right. We agree that this is relevant information to be added. It is well known that at certain timepoints during he pandemic, due to a shortage of reagents, many labs in the world carried out the diagnostic RT-qPCR assay starting from crude lysate of UTM instead of by purifying RNA. There is a linear correlation between the CT obtained from purified RNA and from UTM lysate; besides the literature cited by the reviewer our group also addressed this issue (Brandolini et al., 2021 doi: 10.3390/v130610229) Indeed, the lab has verified the use of crude lysate and a mean of 5 Cts of difference was observed. However, in 2020 all UTM samples for Covid-19 diagnostic procedures had undergone a RNA purification step before diagnostic PCR (only purified RNA was used as a template for RT-qPCR). This was more clearly stated in the revised manuscript on line 129 of the revised manuscript. Also, among the 51 samples used for establishing and evaluating the linear regression models most of them had been assayed with methods different from Seegene but all of them have been re-analysed using Seegene. Hence, since the linear regression function is valid only for the Seegene method, the cohort of ~ 6000 Covid-19 cases was selected among those diagnosed with Seegene. In this regard a new Figure has been added to the revised manuscript (Figure 1new) Finally, the details of the Seegene method were reported in the supplementary materials since it was considered not so relevant and since it is one of the most used Sars-CoV-2 diagnostic assays in the world. Indeed, the cohort of ~6000 samples used Ct obtained with the same method of qPCR and purification (criteria of inclusion obviously) among hundreds of thousands of samples analyzed by The Great Romagna Hub Laboratory of Pievesestina, Italy with a multitude of platforms. 

5. I am not sure what is the rational of performing dPCR only once as described in line 161. The authors should perform multiple times to see the consistency. Related to this, it appears that the authors performed the regression with a single training and test split, but it would be better to perform k-fold cross validation with multiple restarts (meaning randomly separate all samples to training and test sets many times). This would help to understand if their model is generalizable or specific to the particular training/test (evaluation) sets. Additionally, the percentage of data in the training and evaluation sets differs between the linear and logistic regression models.

Concerning the use of a sole dPCR in the evaluation set compared with the triplicate dPCR reactions used in the training set, the rationale is that the model should be as accurate as possible and hence dPCR was carried out in replicate while the evaluation set served to evaluate the error. Hence, we have evaluated the error when the sample is assessed once by dPCR, likely the error would be lower if the actual VL would have been assessed in triplicate also in the evaluation set. Also consider that the precision of the dPCR data is much better than the error of the Ct values. In this regard the CV results have been reported in the Suppl (Table reporting the CV% across the technical replicates) Material. Hence, using dPCR replicates also for the evaluation sets would have improved only marginally the Error estimation. Since in our opinion the measured errors of the model were good enough, we eventually decided to save resources. 

Concerning the need for using iterative modelling, this would be a meaningful suggestion for regression modelling other than linear ones built on small cohorts. Indeed, establishing an equation to explain a so clear linear relationship between two factors linked by a so close relationship would be almost of no benefit. Indeed, the reliability of the linear model can be fully evaluated by the findings of the evaluation set and, most importantly, by the findings of the predictive (logistic) models. However, the suggestion would be warranted when a small cohort is examined. By using a cohort of more than 6000 samples and after having looked at the results it is very evident that the findings are solid and reproducible. Please, the reviewer should cite some manuscripts which used such a design with so consistent cohorts. 

However, in the revised manuscript we have added as Suppl. Material the entire anonymized dataset including the findings of the predictive models. The last show the consistency of the prediction across different models and probability thresholds.

6. The authors need to show more data instead of just reporting the results of statistical tests or analyses by numbers or table format. This manuscript is not reader-friendly partly due to the lack of plots for most of their key analyses. Individual points will be pointed out below. The better visualization is also important to evaluate their key results such as the performance of linear and logistic regression.

This is quite an original concern. What does it means “just reporting the results of the statistical tests or analysis by number of table formats”? We examined 6208 samples by means of logistic regression. We reported the Table summarizing the dataset (cohorts), the equations of logistic regression in a way that anyone can apply the equation to its own dataset and verify the effectiveness and the results of the analysis. These results need to be discussed. Also, plots are tools to summarize and report data. However, to meet the suggestion of the reviewer we have added 6 new Figures reporting plots and graphs (also more than one graph per figure). Since the Plots of the software used to analyse data (STATA) are not reader-friendly we used instead another software (Analyse-it Software, Analyse-it Software, Ltd. UK)), This information has been added to the M&M section and where appropriate.

We would also stress that the most useful results of this study are the functions itself embedded in a spreadsheet. So also the excel file embedding the formula has been added as suppl. Material in the revised manuscript. We are confident that the reviewer will appreciate the revised manuscript as more reader-friendly.

7. The authors claimed there was no significant difference between retested and original Ct values for 51 samples they analyzed. They need to show a plot showing the relationship between retested and original values for both SARS-CoV-2 genes and control genes. It is important to show how consistent these values are to evaluate their results.

Done accordingly

8. The authors need to show plots showing Ct values and log10(measured VLs) with a regression line for 51 samples they analyzed in addition to table 2. This will help readers to evaluate their model. Based on the table, the difference between measured and calculated VLs is not small even though they their regression was significant, and it is hard to understand the pattern of distribution without a plot. Additionally, the R-squared value (0.900) on the training set is quite low, especially for data that is distributed across multiple log-orders. Likewise, the MAD on the evaluation data is >50%, with some samples being off by up to 10-fold, suggesting that the model does not generalize well.

We have added the Figures reporting the plots, accordingly. We respectfully do not agree with the reviewer’s claim in the last paragraph of point 8. “. Evidently, the reviewer is not used to Ct data. Please, note that a difference of 3.3 CT difference means a difference of 10 times. We included in the model many samples with low Cts and a huge difference. Also, it is a bit surprising the claim that a 0.900 R-squared value is quite low, In our opinion 0.900 is remarkable since it is not the result of a serially diluted series but of a correlation between two different measurements. Additionally, in the revised manuscript as requested by the reviewer we have also added a further Figure which should ease the interpretation of the linear regression model, representing that the linear model obtained an outstanding 0.918 R-squared value in the evaluation set. Hopefully, this further evidence should clarify the concerns of the reviewer.

9. The results of logistic regression should be also presented by some plots instead of just showing numbers as a table. The methods also describe using the VL as both a continuous and categorical variable, but only the results from the categorized version are shown in table 4. The authors need to show both results. It would also be helpful to show key values such as sensitivity as a function of probability threshold. Even though the authors used three thresholds, using more thresholds and showing them as plots would be more helpful.

We do understand that the reviewer would prefer plots to numbers. So, we added many plot graphs to the revised manuscript. Hopefully this should increase the readability of the manuscript. 

Concerning the need to show both continuous and categorical VL variables the reviewer does not explain the reasons. For sure, when evaluating the logistic model, one tries a multitude of models using a step forward or step backward method and using also redundant variables as the same predictor entered either as continuous or categorical. Clearly the use of a categorial variable has the advantage of being simpler and more intuitive to infer the risk (odds ratio) while continuous variables are more easy to transfer to the predictor equation. As a matter of fact, the categorization of cVL allowed to easily interpret the odds ratio of each category. The criteria used to drive the choice of the model is the best R-square of the overall model. 

Finally, we also added a plot of the ROC curve analysis showing the optimal threshold while in the revised manuscript the different models have been compared using Area under ROC curves.

Minor comments

1. It would be more helpful for readers to generate a figure explaining the experimental layout, which is nicely described by text in the Materials and methods section.

We thank the reviewer for this suggestion but eventually decided to give up since the revised manuscript has already 9 Figures.

2. In page 10 (line 181), the authors cited “Supp. Mat” though I was not able to find corresponding description or data in the attached supplementary materials.

We regret this missing information. In the revised manuscript an excel file including the equations to calculate the cVL and the probability of death has been added.

3. Figure 1 needs a label saying “calculated viral load (copies/µl)”. 

Done as suggested including this information in the legend (new Figure 5).

4. Figure 2 was not properly labeled. The authors need to show the axis labels and what black curve and grey histogram indicate. Although the text mention about 90th and 95th percentile by citing this figure, it doesn’t look those values are properly presented in this figure. I think it looks a lot better to have 2 stacked plots instead of current partially overlaid one.

Figure 2 (renamed Figure 6 in the revised manuscript) has been re-labelled. Furthermore, the caption has been re-phrased to better represent what is reported in more detail. Also, a graph representing the 90th and 95th percentiles has been added in the same Figure to accurately indicate the results. Thanks for the suggestion. The format has been left overlapped instead of stacked for our personal preferences. (Please, consider that the straight line has been reported as such by the National Health authorities as cited and we do not have the raw data that generated such graph. So minimal adjustment could be made.

5. Infectivity threshold at 1500 copies per µl sounds somewhat arbitrary. The authors need to cite references or explain more.

The appropriate references and the limit of defining this threshold have been acknowledged in the revised manuscript by appropriate re-phrasing on lines 368-371.

6. Although the authors cite Vasudevan et al. 2021, they should also include more discussion about how their results compare to previous efforts to use digital PCR to quantify VL.

Some interesting comparisons are mentioned on lines 355-359 of the revised manuscript. 

7. The authors should explain and justify their sample sizes for their linear and logistic regression models. The original linear regression model is fit on only 13 samples, but then applied to over 3000 samples in the logistic regression model.

DPCR is much more expansive and time-demanding than RT-qPCR. The advantage we wanted to highlight was to build models by correlating dPCR and RT-qPCR in small samples to be applied in large samples. All the findings highlighted in the revised manuscript corroborate such approach and also explain the limits. We have stressed this approach by re-phrasing the aim at the end of the introduction section on lines 108-110.

8. The authors should add original and retested Ct values in table 2.

Done accordingly

9. The authors should provide a supplementary table that include all the information about patients (age, sex, etc) and viral loads (Ct, inferred VL) for ~6000 samples they used.

Done accordingly

6. PLOS authors have the option to publish the peer review history of their article (what does this mean?). If published, this will include your full peer review and any attached files.

Do you want your identity to be public for this peer review? For information about this choice, including consent withdrawal, please see our Privacy Policy.

Reviewer #1: No

On behalf of all the authors

Yours sincerely,

Fabio Gentilini

---

## [Editor Report · Decision Letter 1]

19 Nov 2021

Modelling RT-qPCR cycle-threshold using digital PCR data for implementing SARS-CoV-2 viral load studies

PONE-D-21-12462R1

Dear Dr. Gentilini

We’re pleased to inform you that your manuscript has been judged scientifically suitable for publication and will be formally accepted for publication once it meets all outstanding technical requirements.

Kind regards,

Jean-Luc EPH Darlix, MG, Ph.D.

Academic Editor

PLOS ONE
---

## [Editor Report · Acceptance letter]

1 Dec 2021

PONE-D-21-12462R1 

Modelling RT-qPCR cycle-threshold using digital PCR data for implementing SARS-CoV-2 viral load studies 

Dear Dr. Gentilini:

I'm pleased to inform you that your manuscript has been deemed suitable for publication in PLOS ONE. Congratulations! Your manuscript is now with our production department. 

Kind regards, 

on behalf of

Professor Jean-Luc EPH Darlix 

Academic Editor

PLOS ONE